# Vector Quantized Diffusion Model with CodeUnet for Text-to-Sign Pose Sequences Generation

## Abstract

Sign Language Production (SLP) aims to translate spoken languages into sign sequences automatically. The core process of SLP is to transform sign gloss sequences into their corresponding sign pose sequences (G2P). Most existing G2P models usually perform this conditional long-range generation in an autoregressive manner, which inevitably leads to an accumulation of errors. To address this issue, we propose a vector quantized diffusion method for conditional pose sequences generation, called PoseVQ-Diffusion, which is an iterative non-autoregressive method. Specifically, we first introduce a vector quantized variational autoencoder (Pose-VQVAE) model to represent a pose sequence as a sequence of latent codes. Then we model the latent discrete space by an extension of the recently developed diffusion architecture. To better leverage the spatial-temporal information, we introduce a novel architecture, namely CodeUnet, to generate higher quality pose sequence in the discrete space. Moreover, taking advantage of the learned codes, we develop a novel sequential k-nearest-neighbours method to predict the variable lengths of pose sequences for corresponding gloss sequences. Consequently, compared with the autoregressive G2P models, our model has a faster sampling speed and produces significantly better results. Compared with previous non-autoregressive G2P methods, PoseVQ-Diffusion improves the predicted results with iterative refinements, thus achieving state-of-the-art results on the SLP evaluation benchmark.

## 1 Introduction

Sign Language Production (SLP), as an essential task for the Deaf community, aims to provide continuously sign videos for spoken language sentences. Since sign languages are distinct linguistic systems [1] which differ from natural languages, sign languages have different word orders from their corresponding natural languages. Therefore, directly learning the alignment mapping between them is challenging. To tackle this issue, previous works first translate spoken languages into glosses[1], then generate the sign pose sequences based on the gloss sequences (G2P) [2, 3], and finally optionally use the sign pose sequence to generate the photo-realistic sign video [4]. Accordingly, G2P is the heart procedure of this task, and it is the focus of this paper.

Existing approaches for G2P can be categorized into autoregressive [2, 3] and non-autoregressive [5] methods depending on their decoding strategies. Autoregressive models [2, 3] generate the next pose frame depending on previous frames relying on the teacher forcing strategy [6]. In inference, the recurrent decoding is likely to lead to prediction error propagation over time due to the exposure bias [7]. To break the bottleneck of autoregression, non-autoregressive methods are proposed to

---

[1]Sign glosses are spoken language words that match the meaning of signs and, linguistically, manifest as minimal lexical items.

Submitted to 36th Conference on Neural Information Processing Systems (NeurIPS 2022). Do not distribute.

induce the decoder to generate all target predictions simultaneously [8, 9]. Huang *at al.* [5] proposed a non-autoregressive G2P model to generate sign pose sequence parallelly in a one-shot decoding scheme, and used an External Aligner (EA) for sequence alignment learning.

Motivated by the recent developed Discrete Denoising Diffusion Probabilistic Model (D3PMs) [10, 11, 12] which achieved impressive results for language generation and vector quantized image generation. We propose a Pose Vector Quantized Diffusion (PoseVQ-Diffusion) model to learn the conditional pose sequence generation in the latent discrete space instead of the continuous coordinate space. It is also a non-autoregressive method that performs parallel refinement on the generated results with iterations and therefore shows expressive generative capacity.

We will elaborate our approach in three steps. Firstly, we utilize a vector quantized variational autoencoder (VQ-VAE) to represent the pose sequence as sequential latent codes. Different from image VQ-VAE [13, 14], we devise a specific architecture, Pose-VQVAE, to learn the meaningful codebook by reconstructing the pose sequence. Specifically, we divide a sign skeleton into three local point patches representing pose, right hand and left hand separately. Furthermore, a Tokenizer with a vector quantized variational autoencoder is designed to learn discrete point tokens containing local semantic information.

Next, we extend the standard vector quantized diffusion methods [11, 12] to model the sequential alignments between sign glosses and quantized codes. The discrete diffusion model samples the data distribution by reversing a forward diffusion process that gradually corrupts the input via a fixed Markov chain. Its corruption process by adding noise data (*e.g.*, [MASK] token) draws our attention to the mask-based generative model, Mask-Predict [9], which is proved to be a variant of diffusion model [11]. In this paper, we explore two variants of the diffusion model for our quantized pose sequence generation. Expanding further, to better leverage the spatial and temporal information of the quantized pose sequences, we introduce a new architecture CodeUnet. In contrast to Unet [15] which is a "fully convolution network" for image data, CodeUnet is a "fully transformer network" designed for discrete tokens. As a result of iterative refinements and better spatial-temporal modelling, our model achieves a higher quality of the conditional pose sequence generation.

Lastly, the length prediction of the non-autoregressive G2P models is challenging since the corresponding lengths of different sign glosses are different and variable. In this paper, we propose a novel clustering method for this typical sequential data that local adjacent frames should belong to a cluster. Specifically, taking advantage of the meaningful learned codes in the first stage, we firstly apply the k-nearest-neighbor based density peaks clustering algorithm [16, 17] to locate the peaks with higher local density. Secondly, we design a heuristic algorithm to find the boundary between two peaks according to their semantic distance with the two peak codes. Finally, we leverage the length of each gloss as the additional supervised information and predict the length of the gloss sequence in the inference.

Our model significantly improves the generation quality on the challenging RWTH-PHOENIX-WEATHER-2014T [18] dataset. The evaluation of conditional sequential generation is evaluated using a back-translated model. Extensive experiments show that our model increases the WER score from $82.01\%$ [5] to $78.21\%$ on generated pose sequence to gloss sequence, and BLEU score from $6.66$ [5] to $7.42$ on generated pose sequence to spoken language.

## 2 Related Works

**Sign Language Production.** Most sign language works focus on sign language recognition (SLR) and translation (SLT) [18, 19, 20, 21, 22, 23], aiming to translate the video-based sign language into text-based sequences. And few attempts have been made for the more challenging task of sign language production (SLP) [24, 25]. Stoll *et al.* proposed the first deep SLP model, which adopts the three-step pipeline. In the core process for G2P, they learn the mapping between the sign glosses and the skeleton poses via a look-up table. After that, B. Saunders *et al.* [3] proposed the progressive transformer to learn the mapping with an encoder-decoder architecture and generate the sign pose in an autoregressive manner in the inference. Further, B. Saunders *et al.* [4] proposed a Mixture Density Network (MDN) to generate the pose sequences condition on the sign glosses and utilize a GAN-based method [26] to produce the photo-realistic sign language video. B. Saunders *et al.* [27] translates the spoken language to sign language representation with an autoregressive Transformer network, and uses the gloss information to provide the additional supervision. Then they propose a

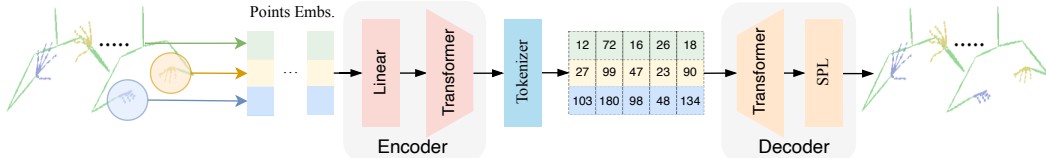

Figure 1: The architecture of the first stage model Pose-VQVAE for learning the discrete latent codes.

Mixture of Motion Primitives(MoMP) architecture to combine distinct motion primitives to produce a continuous sign language sequence.

Different from these autoregressive methods, Huang *et al.* [5] proposed a non-autoregressive model to parallelly generate the sign pose sequence avoiding the error accumulation problem. They apply the monotonic alignment search [28] to generate the alignment lengths of each gloss. Our model also explores a non-autoregressive method with a diffusion strategy, and the adopted diffusion model architecture provides us with a chance to refine the results with multiple iterations.

**Denoising Diffusion Probabilistic Models.** Diffusion generative models have achieved outstanding results on continuous data, such as image generation [29, 30, 31, 32, 33] and speech synthesis [34, 35, 36]. However, most previous works focus on Gaussian diffusion processes that operate in continuous state spaces. The discrete diffusion model is first introduced in [37], and it is applied to text generation in Argmax Flow [10]. To improve and extend the discrete diffusion model, D3PM [11] use a structured categorical corruption process to shape data generation and embed structure in the forward process. VQ-Diffusion [12] apply the discrete diffusion model to conditional vector quantized image synthesis with a mask-and-replace diffusion strategy.

## 3 The Proposed Method

The overall objective of this work is to extend the discrete diffusion model for conditional sign pose sequence generation. The proposed PoseVQ-Diffusion model consists of three key components, Pose-VQVAE to learn the latent codes, a discrete diffusion model with CodeUnet to model the discrete codes generation, and a sequential-KNN algorithm on the length prediction for this non-autoregressive method.

### 3.1 Pose VQ-VAE

In this section, we introduce how to tokenize the points of a sign pose skeleton into a set of discrete tokens. A naive approach is to treat per point as one token. However, such a points-wise reconstruction model tends to tremendous computational cost due to the quadratic complexity of self-attention in Transformers. On the other hand, since the details of hand points are essential for sign pose understanding, treating all the points into one token leads to remarkable inferior reconstruction performance. To achieve a better trade-off between quality and speed, we propose a simple yet efficient implementation that groups the points of a sign skeleton into three local patches, representing pose, right hand and left hand separately. Figure 1 illustrates the framework of our proposed Pose-VQVAE model with the following submodules.

**Encoder.** Given a sign pose sequence of $N$ frames $\mathbf{s} = (s_1, s_2, ..., s_n, ..., s_N) \in \mathbb{R}^{N \times J \times K}$, where $\{x_n^j\}_{j=1}^J$ presents a single sign skeleton containing $J$ joints and $K$ denotes the feature dimension for human joint data. We separate these points into three local pathes, $\mathbf{s}_p \in \mathbb{R}^{N \times (J_p \times K)}$, $\mathbf{s}_r \in \mathbb{R}^{N \times (J_r \times K)}$, $\mathbf{s}_l \in \mathbb{R}^{N \times (J_l \times K)}$ for pose, right hand and left hand respectively, where $J = J_p + J_r + J_l$. In the encoder module $E(e|\mathbf{s})$, we first transform these three points sequences into feature sequences by simple three linear layers and concatenate them together. Then we apply a spatial-temporal Transformer network to learn the long-range interactions within the sequential point features. Finally, we arrive at the encoded features $\{e_n \in \mathbb{R}^{3 \times h}\}_{n=1}^N$.

**Point Tokenizer.** Similar to image VQ-VAE [14], we take the encoded features as inputs and convert them into discrete tokens. Specifically, we perform the nearest neighbors method $\mathcal{Q}(z|e)$ to quantize the point feature to the quantized features $\{z_n \in \mathbb{R}^{3 \times h}\}_{n=1}^N$. The quantized features are maintained by a codebook whose size is $V$.

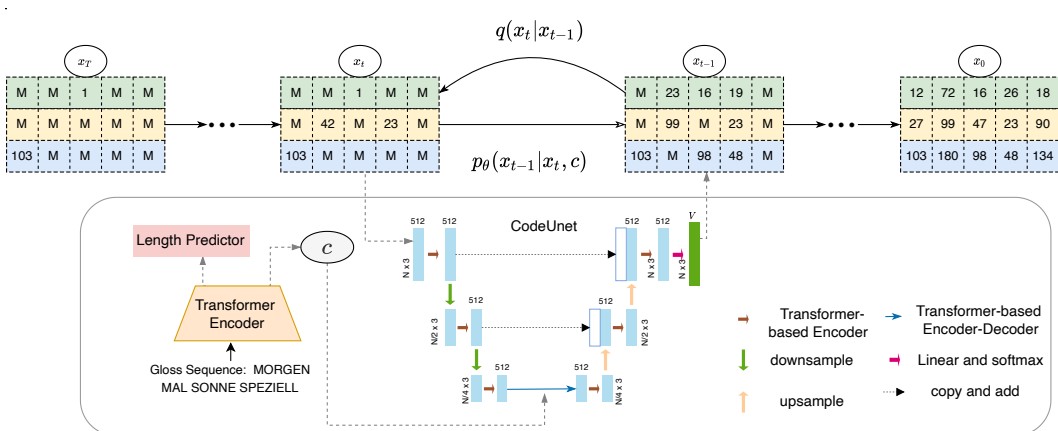

Figure 2: Our approach uses a discrete diffusion model to represent the Vector-Quantized sign pose sequence allowing non-autoregressive pose sequence generation. Specifically, after compressing the sign pose sequences to meaningful discrete codes, each code is randomly masked or replaced and a CodeUnet model is trained to restore the original data.

**Decoder.** The decoder $D(\tilde{\mathbf{s}}|z)$ receives the quantized features as inputs and also applies spatial-temporal Transformer to get the output features $\{o_n \in \mathbb{R}^{3 \times h}\}_{n=1}^{N}$. Finally, we separate the output feature for three sub-skeleton and utilize a structured prediction layer (SPL) [38] $\mathcal{P}(\tilde{s}|o)$ to reconstruct the corresponding sub-skeleton $\tilde{\mathbf{s}}_p \in \mathbb{R}^{N \times (J_r \times K)}$, $\tilde{\mathbf{s}}_l \in \mathbb{R}^{N \times (J_r \times K)}$, $\tilde{\mathbf{s}}_r \in \mathbb{R}^{N \times (J_r \times K)}$. We adopt the SPL to rebuild the skeleton from feature because it explicitly model the spatial structure of the human skeleton and the spatial dependencies between joints. The hierarchy chains of the pose, right hand and left hand skeleton are given in supplemental material.

**Training.** The encoder $E(e|\mathbf{s})$, tokenizer $\mathcal{Q}(z|e)$ and decoder $D(\tilde{\mathbf{s}}|z)$ can be trained end-to-end via the following loss function:

$$\mathcal{L}_{\text{Pose-VQVAE}} = ||\mathbf{s}_p - \tilde{\mathbf{s}}_p|| + ||\mathbf{s}_r - \tilde{\mathbf{s}}_r|| + ||\mathbf{s}_l - \tilde{\mathbf{s}}_l|| + ||sg[e] - z|| + \beta||sg[z] - e||, \tag{1}$$

where $sg[\cdot]$ stands for stop-gradient operation. In practice, we replace the forth term with exponential moving averages (EMA) to update the codebook.

## 3.2 Discrete Diffusion Model with CodeUnet

To allow conditional sampling, a discrete diffusion model is trained on the latent codes obtained from the Pose-VQVAE model. Figure 2 shows the architecture of our proposed PoseVQ-Diffusion, which aims to model the latent space in a non-autoregressive manner. We will subsequently introduce the diffusion process, reverse denoising process and the parametered model CodeUnet.

**Diffusion Process.** Given a sequence of latent codes $x_0 \in \mathbb{R}^{N \times 3}$ obtained from the vector quantized model, where $x_0^{(i,j)} \in \{1, 2, ..., V\}$ at location $(i, j)$ represents the index within the codebook. The diffusion process aims to corrupt the original data $x_0$ via a fixed Markov chain $p(x_t|x_{t-1})$ by adding small amount noise continuously. After a fixed $T$ timesteps, it produces a sequence of increasingly noisy data $x_1, .., x_T$ with the same dimensions as $x_0$, and $x_T$ becomes a pure noise sample.

For the scalar discrete variables with $V$ categories $x_t^{(i,j)} \in [1, V]$, the forward transition probabilities from $x_{t-1}$ to $x_t$ can be represented by matrices $[Q_t]_{mn} = q(x_t = m|x_{t-1} = n) \in \mathbb{R}^{V \times V}$. Note that we omit the superscripts $(i, j)$ to avoid confusion. Then the forward diffusion process can be written as:

$$q(x_t|x_{t-1}) = \mathbf{x}_t^T Q_t \mathbf{x}_{t-1}, \tag{2}$$

where $\mathbf{x}_t \in \mathbb{R}^{V \times 1}$ is the one-hot version of $x_t$ and $Q_t \mathbf{x}_{t-1}$ is the categorical distribution for $x_t$. A nice property of the above Markov diffusion process is that we can sample $x_t$ as any timestep directly from $x_0$ as:

$$q(x_t|x_0) = \mathbf{x}_t^T \bar{Q}_t \mathbf{x}_0, \text{with } \bar{Q}_t = Q_t \ldots Q_1. \tag{3}$$

D3PM [11] formulate the transition matrix $Q_t \in \mathbb{R}^{V \times V}$ by introducing a small number of uniform noises to the categorical distribution. As formulated as the first matrix in Eq. (4) with $\alpha \in [0, 1]$ and $\beta_t = (1 - \alpha_t)/V$. It can be interpreted as each token having a probability of $\alpha_t + \beta_t$ to remain the previous value and a probability of $\beta_t$ to be the value from the whole $V$ categories. Based on D3PM, VQ-Diffusion [12] propose a mask-and-replace diffusion strategy that not only replaces the previous value but also insert [MASK] token to explicitly figure out the tokens that have been replaced. We extend this mask-and-replace strategy to our quantized pose sequence modelling. Since the length of pose sequences may be different in a minibatch, we have to add two special tokens, [MASK] and [PAD] tokens, so each token has $V + 2$ states. The mask-and-replace diffusion process can be defined as follows: each token has a probability of $\alpha_t$ to be unchanged, $V\beta_t$ to be uniformly resampled and $\gamma_t = 1 - \alpha_t - V\beta_t$ to be replaced with [MASK] token. Note that [MASK] and [PAD] tokens always keep its own state. The difference is that [PAD] is used to represent the padding part in the initial sequence $x_0$, and [MASK] is used to replace the original token of the code sequence in the diffusion process. Moreover, in the revised denoising process, the [MASK] positions are required to predict but [PAD] positions need to be ignored. The transition matrix $Q_t \in \mathbb{R}^{(V+2)\times(V+2)}$ is formulated as the second matrix of the following:

$$
Q_t = \begin{bmatrix} \alpha_t + \beta_t & \beta_t & \cdots & \beta_t \\ \beta_t & \alpha_t + \beta_t & \cdots & \beta_t \\ \vdots & \vdots & \ddots & \vdots \\ \beta_t & \beta_t & \cdots & \alpha_t + \beta_t \end{bmatrix}; Q_t = \begin{bmatrix} \alpha_t + \beta_t & \beta_t & \cdots & \beta_t & 0 & 0 \\ \beta_t & \alpha_t + \beta_t & \cdots & \beta_t & 0 & 0 \\ \vdots & \vdots & \ddots & \vdots & \vdots & \vdots \\ \beta_t & \beta_t & \cdots & \alpha_t + \beta_t & 0 & 0 \\ \gamma_t & \gamma_t & \cdots & \gamma_t & 1 & 0 \\ 0 & 0 & \cdots & 0 & 0 & 1 \end{bmatrix}. \tag{4}
$$

Finally, the categorical distribution of $\mathbf{x}_t$ can be derived as following using reparameterization trick:

$$
\text{when } x_0 \neq V + 2, \quad \bar{Q}_t \mathbf{x}_0 = \begin{cases} \bar{\alpha}_t + \bar{\beta}_t, & x_t = x_0 \\ \bar{\beta}_t, & x_t \neq x_0 \text{ and } x_t \leq V \\ \bar{\gamma}_t, & x_t = V + 1 \\ 0, & x_t = V + 2 \end{cases}
$$

$$
\text{when } x_0 = V + 2, \quad \bar{Q}_t \mathbf{x}_0 = \begin{cases} 0, & x_t \neq V + 2 \\ 1, & x_t = V + 2 \end{cases} \tag{5}
$$

where $\bar{\alpha}_t = \prod_{i=1}^{t} \alpha_i$, $\bar{\gamma}_t = 1 - \prod_{i=1}^{t}(1 - \gamma_i)$, and $\bar{\beta}_t = (1 - \bar{\alpha}_t - \bar{\gamma}_t)/V$. Therefore, we can directly sample $x_t$ within the computation cost $O(V)$.

**Reverse Denoising Process.** The reverse denoising process aims to recreate the real sample from a full noise input by gradually sampling from $q(x_{t-1}|x_t)$. However, it is intractable to estimate the conditional probability $q(x_{t-1}|x_t)$ since it needs to use the whole dataset. Fortunately, the conditional probability is tractable when conditioned on $x_0$ using Bayes' rule:

$$
q(x_{t-1}|x_t, x_0) = \frac{q(x_t|x_{t-1}, x_0)q(x_{t-1}|x_0)}{q(x_t|x_0)} = \frac{(\mathbf{x}_t^T Q_t \mathbf{x}_{t-1})(\mathbf{x}_{t-1}\bar{Q}_{t-1}\mathbf{x}_0)}{\mathbf{x}_t^T \bar{Q}_t \mathbf{x}_0}, \tag{6}
$$

thus we train a denoising model $p_\theta(x_{t-1}|x_t, c)$ to approximate the tractable distribution $q(x_{t-1}|x_t, x_0)$, where $c$ is the conditional feature of gloss sequence. And the model is trained to minimize the variational lower bound [37]:

$$
\mathcal{L}_{vb} = \mathbb{E}_q[\underbrace{D_{\text{KL}}(q(x_T|x_0) \parallel p_\theta(x_T))}_{L_T} + \sum_{t=2}^{T} \underbrace{D_{\text{KL}}(q(x_{t-1}|x_t, x_0) \parallel p_\theta(x_{t-1}|x_t, c))}_{L_{t-1}}
$$
$$
- \underbrace{\log p_\theta(x_0|x_1, c)}_{L_0}]. \tag{7}
$$

**Reparameterization Trick on Model Learning.** Compared with directly predicting $p_\theta(x_{t-1}|x_t, c)$, recent works [30, 10, 12] find that predict the data $x_0$ gives better quality at every reverse step. Thus,

we let our denoising model to predict the distribution $p_\theta(\tilde{x}_0|x_t, c)$. With a reparameterization trick, the conditional reverse distribution can be formulated as:

$$p_\theta(x_{t-1}|x_t, c) = \sum_{\tilde{x}_0=1}^{V} q(x_{t-1}|x_t, \tilde{x}_0)p_\theta(\tilde{x}_0|x_t, y). \tag{8}$$

Under this $x_0$-parameterization trick, we introduce an auxiliary denoising objective to encourage good predictions of the data $x_0$ at each time step [11]. The final loss function is combined with the negative variational lower bound and the auxiliary loss:

$$\mathcal{L}_{ddm} = \mathcal{L}_{vb} - \lambda \, log p_\theta(x_0|x_t, c), \tag{9}$$

where $\lambda$ is a coefficient for the auxiliary loss term.

**CodeUnet for Model Learning.** Most image diffusion models [29, 30, 39] adopt the Unet [15] as their architectures since it is effective for data with spatial structure. However, directly applying the Unet in discrete sequence generation, *e.g.*, text generation [11] and quantized image synthesis [12], will bring information leakage problem, since the convolution layer over adjacent tokens may provide shortcuts for the mask-based prediction [40]. Therefore, Austin *et al.* [11] and Gu *et al* [12] use the token-wise Transformer framework to learn the distribution $p_\theta(\tilde{x}_0|x_t, c)$. In this work, to incorporate the advantages of Unet and Transformer networks, we propose a novel architecture CodeUnet to learn the spatial-temporal interaction for our quantized pose sequence generation.

As shown in Figure 2, the CodeUnet consists of a contracting path (left side), an expansive path (right side) and a middle module. The middle module is an encoder-decoder Transformer framework. The encoder consists of 6 Transformer blocks. It takes the gloss sentence as input and obtains a conditional feature sequence. The decoder has two blocks. Each block has a self-attention, a cross-attention, a feed-forward network and an Adaptive Layer Normalization [41, 12](AdaLN). The AdaLN operator is devised to incorporate timestep $t$ information as $AdaLN(h, t) = \alpha_t LayerNorm(h) + b_t$, where $h$ is the intermediate activations, $\alpha_t$ and $\beta_t$ are obtained from a linear projection of the timestep embedding.

Both contracting path and the expansive path are hierarchical structures and each level has two Transformer encoder blocks. For downsampling in contracting path, given the feature of quantized pose sequence, *e.g.*, $h \in \mathbb{R}^{N \times 3 \times d_{model}}$, where $d_{model}$ is the feature dimension, we first sample uniformly with stride 2 in the temporal dimension and remain constant in the spatial dimension. Then we set the downsampled feature as query $Q \in \mathbb{R}^{N/2 \times 3 \times d_{model}}$, and keep key $K$ and value $V$ unchanged for the following attention network. In the upsampling of expansive path, we directly repeat the feature 2 times as a query, but the key and value remains for the following attention network:

$$\forall n = 1, ..., N, Q_n^{up} = h_{n//2}, K^{up} = V^{up} = h, \tag{10}$$

where $\cdot//\cdot$ denotes floor division. Finally, a linear layer and a softmax layer are applied to make the prediction.

### 3.3 Length Prediction with Sequential-KNN

In this section, inspired by [17] which merges tokens with similar semantic meanings from different locations, we propose a novel clustering algorithm to get the lengths for corresponding glosses. Specifically, given a token sequence which is obtained from the Pose-VQVAE model, we compute the local density $\rho$ of each token according to its k-nearest-neighbors:

$$\rho_i = exp(-\frac{1}{k} \sum_{z_j \in KNN(z_i)} \|z_i - z_j\|_2^2), \text{ where } |i - j| <= l \tag{11}$$

where $i, j$ is the position in the sequence, $l$ is a predefined hyperparameter indicating that we only consider the local region since the adjacent tokens are more likely to belong to a gloss. $z_i$ and $z_j$ are the latent feature for $i^{th}$ and $j^{th}$ tokens. $KNN(x_i)$ represents the k-nearest neighbors for $i^{th}$ token.

We assign $\{p_1, ..., p_M\}$ positions with a higher local density as the peaks, where $M$ is the length of the gloss sequence. Then between two adjacent peaks, for example $p_1$ and $p_2$, we sequentially iterate from $p_1$ to $p_2$, and find the first position that is farther from $z_{p_1}$ and closer to $z_{p_2}$, which is

the boundary we determined. After finding these boundaries, we get the lengths of the contiguous pose sequence for its corresponding glosses. As shown in Figure 2, we define the obtained lengths as $\{L_1, .., L_M\}$, and the Transformer encoder for gloss sequence is trained under the supervised information of lengths. For each gloss word, we predict a number from $[1, P]$, where $P$ is the maximum length of the target pose sequence. Mathematically, we formulate the classification loss of length prediction as:

$$\mathcal{L}_{\text{len}} = \frac{\delta}{M} \sum_i^M \sum_j^P (-L_i = j) \log p(L_i|c). \tag{12}$$

In the training of the discrete diffusion mode, $\mathcal{L}_{\text{len}}$ is trained together with a coefficient $\delta$. In the inference, we predict the length of glosses, and their summation is the length of target pose sequence.

In summary, we arrive at our proposed two-stage approach, PoseVQ-Diffusion, with the first-stage Pose-VQVAE model and the second-stage discrete diffusion model with a length predictor. The whole training and inference algorithm is shown in supplementary material.

## 4 Experiments

### 4.1 Experiment Setups and Implementation Details

**Dataset.** We evaluate our G2P model on RWTH-PHOENIX-Weather 2014T (RPWT) dataset [18]. It is the *only* publicly available SLP dataset with parallel sign language videos, gloss annotations, and spoken language translations. This corpus contains 7096 training samples (with 1066 different sign glosses in gloss annotations and 2887 words in German spoken language translations), 519 validation samples, and 642 test samples.

**Evaluation Criteria.** Following the widely-used setting in SLP [3], we adopt the back-translation method for evaluation. Specifically, we utilize the state-of-the-art SLT [19] model to translate the generated sign pose sequence back to gloss sequence and spoken language, where its input is modified as pose sequence. Specifically, we compute BLEU [42] and Word Error Rate (WER) between the back-translated spoken language translations and gloss recognition results with ground truth spoken language and gloss sequence.

**Data Processing.** Since the RWTH-PHOENIX-Weather 2014T (RPWT) dataset doesn't contain pose information, we generate the pose sequence as the ground truth. Following B. Saunders *et al.* [3], we extract 2D joint points from sign video using OpenPose [43] and lift the 2D joints to 3D with a skeletal model estimation improvement method [44]. Finally, similar to [24], we apply skeleton normalization to remove the skeleton size difference between different signers.

**Model Setting.** The Pose-VQVAE consists of an Encoder, a Tokenizer, and a Decoder. The Encoder contains a linear layer to transform pose points to hidden feature with dimension set as 256, a 3-layer Transformer module with divided space-time attention [45]. The Tokenizer maintains a codebook with a size set as 2048. The Decoder contains the same 3-layer Transformer module as the encoder and an SPL layer to predict the structural sign skeleton. For the discrete diffusion model, we set the timestep $T$ as 100. All Transformer blocks of CodeUnet have $d_{\text{model}}$=512 and $N_{\text{depth}}$=2. The size of local region $l$ in Equation 11, is set as 16 which is the average length of a gloss. And the number of nearest neighbors $k$ is set as 16. We optimize our network using AdamW [46] with $\beta_1 = 0.9$ and $\beta_2 = 0.96$. The learning rate is set to 0.0004 after 5000 iterations of warmup. We train the model on 8 NVIDIA Tesla V100 GPUs. We include all hyperparameters setting and the details of implementation in the supplementary material.

### 4.2 Comparisons with State-of-the-Art Methods

**Competing Methods.** We compare our PoseVQ-Diffusion with previous state-of-the-art G2P models. **Progressive Transformer** (PTR) [3] is the first SLP model to tackle the G2P problem in an autoregressive manner. Since they use the ground-truth first sign pose frame and timing information, their reported results are not comparable to ours. Thus we adopt the results reported by Huang *et al.* [5]. **NAT-EA** [5] propose a non-autoregressive method to directly predict the target pose sequence

Table 1: Quantitative results for G2P task on RWTH-PHOENIX-Weather 2014T test dataset. † indicates the results is provided by Huang et al. [5]. Note that smaller WER is better, higher BLEU is better, and lower DTW-MJE is better. The closer all the results are to the GT, the better.

| Method | WER | BLEU-1 | BLEU-2 | BLEU-3 | BLEU-4 | DTW-MJE |
|---|---|---|---|---|---|---|
| PTR† [3] | 94.65 | 11.45 | 7.08 | 5.08 | 4.04 | 0.191 |
| MoMP† [27] | 92.41 | 13.17 | 8.24 | 6.25 | 4.75 | 0.188 |
| NAT-AT [5] | 88.15 | 14.26 | 9.93 | 7.11 | 5.53 | 0.177 |
| NAT-EA [5] | 82.01 | 15.12 | 10.45 | 7.99 | 6.66 | 0.146 |
| **PoseVQ-AR (Ours)** | 85.27 | 14.26 | 10.02 | 7.57 | 5.94 | 0.172 |
| **PoseVQ-MP (Ours)** | 79.38 | 15.43 | 10.69 | 8.26 | 6.98 | 0.146 |
| **PoseVQ-Diffusion (Ours)** | **78.21** | **16.03** | **11.32** | **9.17** | **7.42** | **0.122** |
| GT | 50.23 | 23.47 | 15.86 | 12.03 | 10.47 | 0.0 |

with the External Aligner (EA) to learn alignments between glosses and pose sequence. **NAT-AT** is the NAT model without EA that uses the decoder-to-encoder attention to learn the alignments.

**Quantitative Comparison.** The comparison between our PoseVQ-Diffusion and the competing methods is shown in Tabel 1. The row of **PoseVQ-AR** refs to the vector quantized model with an autoregressive decoder. The row of **PoseVQ-MP** refs to the vector quantized model with the Mask-Predict [9] strategy, which is also a variant of discrete diffusion model [11]. **PoseVQ-Diffusion** refs to the vector quantized model with mask-and-replace diffusion strategy. As indicated in Table 1, both diffusion-based models outperform the state-of-the-art G2P models with relative improvements on WER score by $4.6\%$ ($82.01 \rightarrow 78.21$) and on BLEU-4 by $11.4\%$ ($6.66 \rightarrow 7.42$). This shows the effectiveness of the iterative mask-based non-autoregressive method on the vector quantized pose sequence. In addition, the Mask-Predict strategy is a mask only strategy that is similar to PoseVQ-Diffusion with $\bar{\gamma}_T = 1$. Therefore, PoseVQ-Diffusion achieves better performance than PoseVQ-MP. This reflects the mask-and-replace strategy is superior to the mask only strategy.

### 4.3 Model Analysis and Discussions

We also investigate the effects of different components and design choices of our proposed model.

**Analysis of The Design of Pose-VQVAE.** As shown in the first three rows of Table 2a, we study the design of our Pose-VQVAE model. Pose-VQVAE-joint means we compress all points into one token, Pose-VQVAE-separate means the points are separated into three local patches according to the structure of a sign skeleton. Empirically, Pose-VQVAE-separate achieves much better reconstruction (MSE) performance. This indicates that compressing all skeleton points into one token embedding is not advisable, leading to information loss. The second row of Figure 3 shows the sample of sign pose sequences reconstructed by Pose-VQVAE-separate.

**CodeUnet vs. Transformer.** For a fair comparison, we replace our CodeUnet with Transformer network with keeping other settings the same. As shown in the last three rows of Table 2a, the diffusion-based model with our CodeUnet achieves better performance on the back-translate evaluation. This phenomenon suggests that the hierarchical structure of CodeUnet makes it particularly effective for data with spatial structure. Moreover, in our experiments with the same batch size, CodeUnet coverages faster than Transformer. Having said that, due to sign pose sequences being temporally redundant, the compression of CodeUnet in the time dimension makes it more efficient in training.

**Number of Timesteps.** We compare the performance of the model with different numbers of training steps. As shown in the left two columns of Table 2b, we find that the results get better when the training step size is increased from 10 to 100. As it increased further, the results seemed to saturate. Therefore, we set the training step to 100 to trade off performance and speed.

**Length Candidates.** Length prediction is essential for a non-autoregressive generation. Our approach proposes a Sequential-KNN algorithm to learn the lengths for corresponding glosses and then treat the length prediction as a classification problem. As shown in the right two columns in Table 2b, we study the performance with different length candidates and compare it with the reference (gold) target sequence length. The results show that multiple candidates can increase performance, but too many candidates can even degrade performance.

Sign Gloss: BESONDERS OST DEUTSCH LAND MEHR REGEN ODER SCHNEE

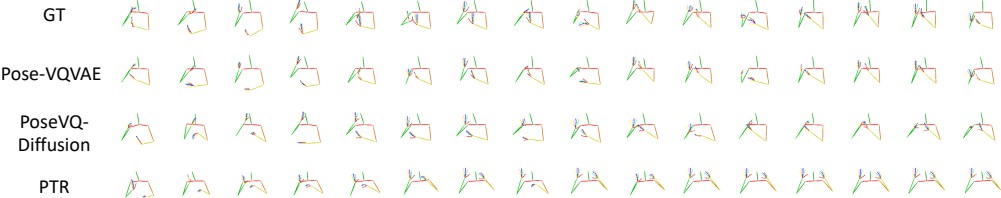

Sign Gloss: DONNERSTAG SUEDOST WEITER WECHSELHAFT NORDWEST MEHR FREUNDLICH SONNE

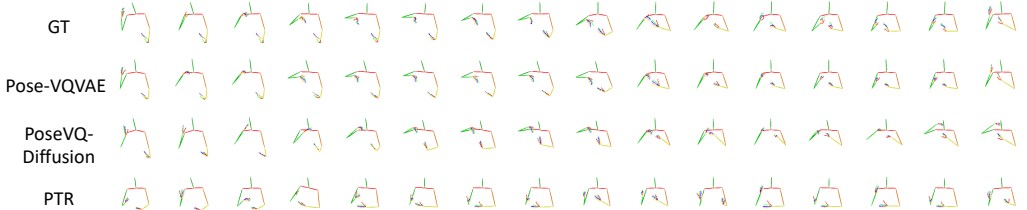

Figure 3: G2P qualitative results. We show some examples of predicted sign pose sequences compared with our reconstruction model and previous G2P model [3]. For readability, we sampled every 5 frames for a total of 16 frames. See our supplementary material for more results.

Table 2: Analysis into the effects of different designs and hyperparameters for our propose model.

(a) Ablation on design of reconstruction and prediction model.

| Reconstruction Model | MSE ($\downarrow$) |
|---|---|
| Pose-VQVAE-joint | 0.0242 |
| Pose-VQVAE-seperate | **0.0139** |

| Prediction Model | WER ($\downarrow$) |
|---|---|
| Transformer | 80.36 |
| CodeUnet | **78.21** |

(b) Ablation on the hyperparameters of training steps and length candidates.

| Training Steps | WER ($\downarrow$) | Length Candidates | WER ($\downarrow$) |
|---|---|---|---|
| 10 | 81.06 | 1 | 79.45 |
| 50 | 79.31 | 2 | 78.69 |
| 100 | 78.21 | 3 | 78.21 |
| 150 | 78.17 | 4 | 78.74 |
| 200 | 78.15 | Gold | 77.26 |

## 5   Conclusion

In this paper, we presented a novel paradigm for conditional sign pose sequence generation through an iterative non-autoregressive method. Specifically, we first devise a specific architecture Pose-VQVAE to learn discrete codes by reconstruction. Then we extend the discrete diffusion model to model the sequential alignments between sign glosses and quantized codes. And a "fully transformer" network CodeUnet is proposed for the spatial-temporal information in discrete space. Finally, we propose a sequential-KNN algorithm to learn the length of corresponding glosses and then predict the length as a classification task. Compared with previous state-of-the-art autoregressive and non-autoregressive methods, extensive experiments demonstrate the effectiveness of our proposed PoseVQ-Diffusion framework.

## 6   Broader Impact and Limitations

We develop a general paradigm for conditional pose generation in this paper. We do not foresee any negative ethical/societal impacts at this moment. Although the proposed PoseVQ-Diffusion proves effective in conditional sign pose sequence generation, we notice several limitations of our approach. (i) Although our discrete diffusion model-based model has a faster sampling speed in longer sequence generation than traditional autoregressive models, it is much slower than one-shot non-autoregressive models. (ii) Our proposed two-stage based models are not end-to-end and thus more difficult to train than previous methods. We, therefore, plan to resolve the aforementioned issues and further mitigate the training and inference speed between the one-shot non-autoregressive method.

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
