# OpenReview forum: "Vector Quantized Diffusion Model with CodeUnet for Text-to-Sign Pose Sequences Generation"
_NeurIPS.cc/2022/Conference — NeurIPS 2022 Submitted_

### Official Review · Reviewer_ndEY · 2022-07-08

**Rating:** 5
**Confidence:** 4
**Soundness:** 3 good
**Presentation:** 3 good
**Contribution:** 2 fair

**Summary:**

The authors propose a novel text-to-sign pose sequences generation method. The proposed method is based on the powerful discrete denoising diffusion probabilistic model. It solves the long-range autoregressive generation problem in a non-autoregressive manner. A vector
quantized conditional diffusion model is proposed for the pose generation. The proposed method achieves state-of-the-art results on the SLP evaluation benchmark.

**Questions:**

1. Have the authors tried to use some other models to replace the CodeUnet? Leveraging transformer-based blocks and Unet is interesting. But how well does the network work compared with other widely used networks?
2. Why is the PAD state used and what is the meaning? It seems using MASK state is enough. And this point is not illustrated in the paper. From the transition matrix, no other state will turn to be PAD and PAD is an isolated state. Then why do we still need to talk about whether
$x_t$ equals V+2?
3. Do you have some other experimental results and ablation studies? It seems experiments should be done to verify the effectiveness of many parts of the work, including the architecture, section 3.3, etc.

**Limitations:**

For the checklist, the 6th and 7th "YES" may be misleading.

**Strengths And Weaknesses:**

# Strengths
1. The proposed method uses a novel diffusion-based model and achieves great performance.
2. The authors argue they solve the problem in an auto-regressive way, which may be meaningful in the relative fields.
3. A Pose-VQVAE is used to learn the meaningful codebook and the discrete diffusion model is devised in a fancy way.
4. The CodeUnet is proposed in the diffusion model for better leveraging the spatial and temporal information of the quantized pose sequences.

# Weaknesses
## Main Issue
1. This work is based on the works of some diffusion-based models, especially the CVPR 2022 paper Vector quantized diffusion model for text-to-image synthesis [1]. My biggest concern is that we can not tell how big the contribution of this work is. On the one hand, this work
introduces the VQ-Diffusion method proposed in a text-to-image generation to the G2P task and achieves great performance, which is impressive. On the other hand, compared with the former CVPR work, the method in this paper uses the CodeUnet, which is not a big step, from my perspective, and only proposes some small points, like dividing the joints into 3 subsets. Therefore, it is really hard to judge how big the contribution is.

## Some Minor Issues
1. A typo in line 107.
2. For the discrete diffusion model part, why there is a PAD state? I think the meaning of the existence of PAD is not illustrated clearly in the section.
3. The experiments part may not be sufficient enough. For example, the method tries to divide the joints into 3 subsets. What will the results be like if we use a simpler implementation?
4. In table 1 in the results, how to evaluate the numerical results? How to define better results should be discussed in the paper. Higher or smaller? Or closer to the GT?
5. In the checklist, the authors say that the code is provided. However, this is missing in the supplementary materials.

[1] Vector quantized diffusion model for text-to-image synthesis, Gu et al., CVPR 2022.

---

> ### Author Response · Authors · 2022-08-02
> **Response to reviewer ndEY**
>
> We would like to thank the reviewer for valuable feedback, and acknowledge that "meaningful in the relative fields" and "in a fancy way"; as well as the appreciation of CodeUnet. We address individual comments below.
>
> ### Weaknesses
>
> #### W1: we can not tell how big the contribution of this work is.
>
> - Our work and VQ-Diffusion[1] share a similar idea for conditional generation, but there are also some important differences in the detailed approach:
> 1. VQ-Diffusion[1] is proposed for text-to-image synthesis. While our model is specifically tailored to text-to-pose sequence generation whose generated length is uncertain. Therefore, a special PAD state is necessary to address the problem of different lengths in a minibatch, and the corresponding transition matrix for diffusion process needs to be changed. Meanwhile, a novel length prediction method with sequential-KNN algorithm is proposed to solve the length prediction problem.
> 2. VQ-Diffusion[1] utilizes the classical VQ-VAE model for image reconstruction. While in our method, we propose a novel architecture Pose-VQVAE to learn meaningful codes.  Compared with image VQVAE, pose-VQVAE provides a reliable idea on how to compress joints of skeleton into high-dimensional semantic codes. And in the decoder, the SPL module is used, which can effectively utilize the hierarchical structure of the human skeleton.
> 3. VQ-Diffusion[1] utilizes the classical Transformer model for  generation. While in our method, we propose a novel architecture CodeUnet with the characteristics of "Unet" for discrete code generation, which is demonstrated more effective in Table2(a). We believe the architecture of CodeUnet can be applied to more relative generation tasks.
>
>
> #### W2: A typo in line 107.
>
> - Thanks for pointing out the typo here, we will fix it in the revised paper.
>
>
> #### W3&Q2:  Why is the PAD state used? The difference between MASK state?
>
> - For different conditional gloss sequences, the corresponding pose sequence lengths are also inconsistent. Therefore, in a minibatch, we need to use PAD state to represent padding parts.
> - In the initial $x_0$, PAD is used to represent the padding part. MASK state is used to replace the original token of the code sequence in the diffusion process, and the MASK positions are required to predict in the training of reverse denoising process. Different MASK state, PAD positions need to be ignored in the predicting and calculating of attention weights. Therefore, PAD is different from MASK but necessary.
> - Although PAD remains unchanged, for efficient computation, PAD is a state that has to be considered during matrix transformation, so $x_t$ equal V+2 need to talk about.
>
> #### W4: The method tries to divide the joints into 3 subsets. What will the results be like if we use a simpler implementation?
>
> - In the first three rows of Table 2a, we experimentally compared Pose-VQVAE-joint and Pose-VQVAE-separate, where Pose-VQVAE-joint compresses all joints into one token. In Line 112~114 of revised paper, we discuss a simpler implementation, "A naive approach is to treat per point as one token. However, such a points-wise reconstruction model tends to tremendous computational cost due to the quadratic complexity of self-attention in Transformers." Moreover, this will result in a very long generation sequence in the second stage, thus affecting the generation quality, and treating each joint as a token doesn't make sense intuitively.
>
> #### W5: In table 1 in the results, how to evaluate the numerical results? How to define better results?
>
> - Thanks for pointing this out. In Table 1, smaller WER is better, higher BLEU is better, and lower DTW-MJE is better. The closer all the results are to the GT, the better.  We will add the above description in the revised paper.
>
> #### W6&L1: code is missing in the supplementary materials.
>
> - Thanks for pointing this out. This is our mistake, we will provide the code in the revised supplementary material.
>
>
> ### Question
>
> #### Q1: Have the authors tried to use some other models to replace the CodeUnet? compared with other widely used networks?
>
> - In Table2(a), we make a comparison with the Transformer network. In non-autoregressive discrete sequence generation tasks, Transformer is the most common model structure. For example, in the field of non-autoregressive text generation[2], as far as we know, there is only one Transformer structure. Therefore, we hope our proposed CodeUnet can provide another choice for these fields.  [2] A Survey on Non-Autoregressive Generation for Neural Machine Translation and Beyond, ACL2022
>
> #### Q3: some other ablation studies? verify the effectiveness of section 3.3, etc.
>
> - In  Length Candidates of 4.3 section and the right two columns in Table 2b, we verify the effective of the algorithm of section 3.3. The last line of right rows of Table 2b means using the ground truth target lengths.
> - We will add the ablation study of model size in the revised paper.

---

> > ### Comment · Reviewer_ndEY · 2022-08-08
> > **Response to authors**
> >
> > I thank the authors for the detailed reply. I recommend the authors to highlight the difference between this work and [1] in the final version. Considering the merit and demerit of this paper, I hence maintain my score for this paper.
> >
> > [1] Vector quantized diffusion model for text-to-image synthesis, Gu et al., CVPR 2022.

---

> > > ### Author Response · Authors · 2022-08-08
> > > **Response to reviewer ndEY**
> > >
> > > Thank you very much for your reply, the difference from work [1] is really important, we decided to add some motivations and insights.
> > >
> > > 1. Firstly, the difference of tasks brings about the difference in the structure of the reconstructed model. [1] is for text to image generation. The spatial structure of the image can be encoded and decoded using traditional cnn/vit, and low-dimensional latent semantic codes can be learned. But the special structure of the sign language skeleton makes it impossible to simply use cnn/vit, how to compress it into a low-dimensional space is a problem. Therefore, we propose **pose-vqvae** to explicitly combine pose points according to human body structure, and verify its effectiveness through experiments. We believe this method can be applied in many pose sequence generation tasks in the future.
> > > 2. Secondly, the image generated by [1] is of a fixed size, such as 256x256. And the sign language sequences we generate are of uncertain length, so **[PAD] state, novel transition matrix and length prediction algorithms** are necessary. Based on the semantic codes learned by pose-vqvae, we propose a feature-similarity clustering-based method to predict the generated length.
> > > 3. Lastly, in the second stage of learning from text to latent code sequences, compared with the transformer structure in [1], we proposed a novel **CodeUnet** framework considering the spatial-temporal characteristics of the sign skeleton. Different from [1], which simply flattens all codes into a sequence, the multi-granularity hierarchical structure of codeunet not only considers the temporal redundancy of the sign language sequence, but also considers the spatial characteristics of the sign skeleton which is composed of pose, the left hand and the right hand.
> > >
> > > We will highlight these differences in the final version. Thanks again for making our paper better.

---

### Official Review · Reviewer_Ljyz · 2022-07-11

**Rating:** 5
**Confidence:** 3
**Soundness:** 3 good
**Presentation:** 3 good
**Contribution:** 2 fair

**Summary:**

This paper tackles the problem of Sign Language Production (SLP), which aims to translate spoken languages into sign pose sequences. The proposed pipeline is based on a Vector Quantized Diffusion model.  The key contribution consists of separating a sign skeleton into three local point patches representing pose, right hand, and left hand separately. An extension of standard VQ diffusion method by considering mask token and a new architecture CodeUnet as well as lengths prediction of sign glosses.

The proposed approach demonstrates its effectiveness on RWTH-PHOENIX-WEATHER-2014T.


**Questions:**

Please refer to the weaknesses.

**Limitations:**

No potential negative societal impact.

**Strengths And Weaknesses:**

**Strength**

* [S1]: The paper is well organized, and the approach is well presented and self-contained.

* [S2]: The improvement on RWTH-PHOENIX-WEATHER-2014T is consistent over different metrics.

**Weakness**

* [W1]: The main weakness is the novelty. The core idea of this paper, _i.e._ VQ Diffusion, has been demonstrated to be successful in many generation tasks. Thus it is not surprising that it works on SLP. Learning different codebooks for different groups of joints is not novel as well, which has been used in music to dance generation [a].

* [W2]: The experimental part is a little bit weak, as only one dataset is considered. Although the paper mentioned there is only one dataset publicly available.

* [W3]: In the related work, the paper mentions [4, 26] as works to generate pose sequences from glosses. I am wondering whether there is a particular reason to make a fair comparison to these approaches.

* [W4]: In the case of using 3 codebooks (pose, right hand, and left hand), the reconstruction becomes simpler however the generation might be harder as well. I think it would be more convincing to validate the idea of separating codebooks on generation rather than reconstruction.

**Minor Questions towards Clarification**

* [C1]: Training details are missing, such as optimizer, weight decay, learning rate strategy etc. Also, it is not clear whether the paper leverages exponential moving average to train the codebook, which is a common practice to train VQ-VAE.

* [C2]: Figure 2: Leangth Predictor → Length Predictor

_[a]: Siyao, Li, et al. "Bailando: 3D Dance Generation by Actor-Critic GPT with Choreographic Memory." Proceedings of the IEEE/CVF Conference on Computer Vision and Pattern Recognition. 2022._

---

> ### Author Response · Authors · 2022-08-02
> **Response to reviewer Ljyz**
>
> We would like to thank the reviewer for valuable and detailed feedback. We address individual comments below.
>
> ### Weakness
> #### W1: the novelty is weak, learning different codebooks for different groups of joints is not novel as well[a]
>
> - Our model is specifically tailored to text-to-pose sequence generation, and its length is uncertain compared to the text-to-image generation problem of VQ-Diffusion. Therefore, 1) a special PAD state is necessary to address the problem of different lengths in the batch, and the corresponding transition matrix also needs to be changed. 2) meanwhile, a novel length prediction method with sequential-KNN algorithm is proposed. 3) For the generation of discrete sequences, we propose a codeunet network that is more suitable for sign pose sequence generation than Transformers since its characteristics of "Unet". In future work, the CodeUnet can also be applied to more discrete sequence generation tasks.
>
> - Our Pose-VQVAE model separates the sign skeleton into three local point patches is similar to Bailando[a] which separates the 3D body into upper and lower half bodies. However, there are some differences, such as  we use a special SPL structure on the decoder, which is very helpful to improve the generation. And the first edition of arxiv for Bailando[a] was on March 25th, so they are works of the same period. We will mention this paper in revised version.
>
> #### W2: The experimental part is a little bit weak
>
> - Most of the existing work on SLP is implemented on this dataset, since PHOENIX dataset is the only dataset that provides spoken language, gloss sequence, and corresponding sign video.
>
> #### W3:  fair comparison with related work[4]
>
> - The work of [4] focuses on the SignGAN for pose-to-video, and their gloss-to-pose model is an extension work of PTR[3] which is an important baseline in our paper. In our paper, we focus on the gloss-to-pose task, and make a comparison with PTR[3]. Moreover, this paper is an Arxiv paper and has not been published yet.
>
> #### W4: the effect of using 3 codebooks on generation, I think it would be more convincing to validate the idea of separating codebooks on generation rather than reconstruction.
>
> - Indeed, the number of codes will affect the generation quality. In the first three rows of Table 2a, we demonstrate that Pose-VQVAE-separates separates the skeleton into three patches making great increments. And in the visualization experiment, the reconstruction effect obtained by Pose-VQVAE-joint is very poor, and it is difficult to guarantee the quality of generation in the second stage. We will provide the visualized results of reconstruction in the revised version.
>
> ### Clarification
>
> #### C1: Training details are missing
>
> - Thank you for pointing this out! In our Pose-VQVAE training, we use the exponential moving average to train the codebook. And we will add more training details in Experimental Setup section in the revised paper.
>
> #### C2: Figure 2: Leangth Predictor → Length Predictor
>
> - Thanks for pointing out the typo here, we will fix it in the revised paper.

---

> > ### Author Response · Authors · 2022-08-09
> > **Open discussion**
> >
> > Dear reviewer Ljyz
> >
> > Many thanks for your constructive comments. You are welcome to provide us feedback if any before the open discussion phase ends. We are glad to answer any follow-up questions.
> >
> > Many thanks,
> >
> > Authors

---

### Official Review · Reviewer_vAR2 · 2022-07-14

**Rating:** 5
**Confidence:** 5
**Soundness:** 3 good
**Presentation:** 2 fair
**Contribution:** 3 good

**Summary:**

The paper is dedicated to the sign language production problem. In the paper, based on VQ-Diffusion, the authors propose the PoseVQ-Diffusion model to generate the pose sequence with iterative refinement. The proposed method achieves encouraging performance on PHOENIX dataset.

**Questions:**

1. In Line 16 – 17, the authors claim that the proposed method has a faster sampling speed compared with autoregressive models. However, there is no experimental results to support this argument. As a matter of fact, diffusion models usually have more model parameters and longer training & sampling times.
2. In Line 164 – 165, the summation of the probability to be unchanged, to be uniformly resampled and to be replaced with [MASK] token is not equal to 1.
3. The authors do not present the qualitative comparison with NAT-AT and NAT-EA.


**Limitations:**

I include several constructive suggestions for improvement.
1. The authors should provide the comparison with the missing methods mentioned above.
2. The authors should compare the model parameter and training & sampling time with previous methods.
3. Other types of evaluation, such as user study, will further verify the effectiveness of the proposed method.


**Strengths And Weaknesses:**

Strength:
The paper leverages the VQ-diffusion model to sign language production, which is novel and interesting. The diffusion model is considered to be potential on generative problems. The attempt made in the paper is very meaningful. Meanwhile, the authors propose several additional improvements for VQ-diffusion on SLP, which is also good.

Weakness:
Despite the novel framework, there are some non-trivial concerns on the effectiveness evaluation of the proposed framework.

First, the authors only compare the proposed method with two previous methods, and the latest method compared is NAT-AT/EA from ACM MM 2021. Several important baseline methods, i.e., [1] and [2], are missing.
[1] Mixed SIGNals: Sign Language Production via a Mixture of Motion Primitives. Ben Saunders, et. al. ICCV 2021.
[2] Signing at Scale: Learning to Co-Articulate Signs for Large-Scale Photo-Realistic Sign Language Production. Ben Saunders, et. al. CVPR 2022.

Secondly, the authors claim that the proposed method has a faster sampling speed compared with autoregressive models. However, there is no experimental results to support this argument.

---

> ### Author Response · Authors · 2022-08-02
> **Response to reviewer vAR2**
>
> We would like to thank the reviewer for valuable feedback, and acknowledge that "novel and interesting" and "the attempt made is very meanfuling". We address individual comments below.
>
> ### Weaknesses
>
> #### W1&L1: Lack of comparison with two important baseline methods[1,2]
> - MoMP[1] is an improved work built upon PTR[3] which is an important baseline method in our paper. MoMP first translates the spoken language to sign language representation with an autoregressive Transformer network and uses the gloss information to provide additional supervision. Then it proposes a Mixture of Motion Primitives(MoMP) architecture to combine distinct motion primitives. However, they didn't release their source code, and some details are not mentioned, such as how the first frame on the decoding side is determined, and how to determine the end of the generated sequence. Note that their work is based on PTR[3], they use the ground truth first sign pose frame and timing information at inference, and their presented results are not comparable to ours. We reproduce MoMP based on their released PTR code without extra ground truth information at inference. We provide the reimplemented results below and add the comparison to the paper.
>
> |Methods|WER|BLEU-1|BLEU-2|BLEU-3|BLEU-4|DTW-MJE|
> |-|-|-|-| -|-|-|
> |MoMP[1]|92.41|13.17|8.24|6.25|4.75|0.188|
>
>
> - Ben et al.[2] tackle the SLP task with two stages. First, they translate the spoken language into gloss sequence, and convert the gloss to corresponding sign pose sequence through the established gloss-sign pose sequence dictionary, then they propose a FS-NET network to learn co-articulation to smooth these sequences into continuous sign language sequences. Second, they utilize a SignGAN to generate photo-realistic continuous sign videos. In their Experiment Setup, they collect exhaustive dictionary examples of every DGS sign present in mDGS[4] and PHOENIX14T, trimmed to remove the sign onset and offset. Specifically, they use the additional frame-level gloss annotations of mDGS to build gloss-sign pose sequence dictionary, thus it's unfair to compare their work with ours.
>
> [3] Progressive transformers for end-to-end sign language production, Ben Saunders, et. al. ECCV2021
>
> [4] DGS Corpus & Dicta-Sign: The Hamburg Studio Setup. Hanke et al.  CSLT2010
>
> #### W2&Q1&L2. sampling speed, training time and model parameters compared with autoregressive models
>
> - Thank you for the advice.  We provide the model parameters, training time and sampling speed compared with several baseline methods. We evaluate the throughput of both methods on a V100 GPU with a batch size of 8. As shown in the below Table, We can see that non-autoregressive methods speed up inference compared to autoregressive methods. Specifically, the iterative non-autoregressive methods VQPose-MP and VQPose-Diffusion are 12 times and 3 times that of the AR model, respectively, with better performance.
>
> |Method| Model parameters | Training cost(minutes*epoch) | throughput(imgs/s) |
> |-|-|-|-|
> | AR methods| | | |
> | PTR[3] | 46.8M| 2.0m* 100| 0.08|
> | VQPose-AR(Ours)| 46.8M| 2.2m * 100 | 0.05|
> | NAR Methods || ||
> | NAT-EA[5]| 23.4M| 1.6m* 300 | 1.74|
> | VQPose-MP(Ours)| 57.2M| 2.5m * 300 | 0.96|
> | VQPose-Diffusion(Ours) | 57.2M| 2.5m * 300 | 0.25|
>
>
> ### Questions
> #### Q2: In Line 164 – 165, the summation of the probability to be unchanged, to be uniformly resampled and to be replaced with [MASK] token is not equal to 1.
>
> - Thank you for pointing this out. In Line 164-165, the probability of $\alpha_t + \beta_t$ to be unchanged not only includes not being replaced by other tokens, but also being replaced by itself. Thus there is an overlap probability of $\beta_t$ between $\alpha_t+\beta_t$ and $V\beta_t$. To make the description more accurate, in our revised paper, we will modify the description to be "each token has a probability of $\alpha_t$ to be unchanged, $V\beta_t$ to be uniformly resampled and $\gamma_t=1-\alpha_t - V\beta_t$ to be replaced with [MASK] token."
>
> #### Q3: The authors do not present the qualitative comparison with NAT-AT and NAT-EA.
>
> - Thank you for your advice, we will add corresponding qualitative results in Figure 3 in the revised paper. Unfortunately, we were unable to complete the required experiments in time for the rebuttal, but are committed to doing so for the camera ready.
>
> ### Limitation
> #### L3: Other types of evaluation, such as user study
>
> - Due to the diversity of generation, standard metrics may not fully reflect whether generation results convey the correct meaning or not.  To further demonstrate our performance gains, we ask two participants to compare the predictions of VQPose-Diffusion and PTR[3] on RPWT to ground-truth pose sequence, and choose the semantically more relevant result. The two participants favor 384 (59.81%) and 413 (64.33%)  translations of VQPose-Diffusion over those of PTR[3] out of 642 testing instances. This experiment further demonstrates our significant improvements.

---

> > ### Author Response · Authors · 2022-08-09
> > **Open discussion**
> >
> > Dear reviewer vAR2
> >
> > Many thanks for your constructive comments. You are welcome to provide us feedback if any before the open discussion phase ends. We are glad to answer any follow-up questions.
> >
> > Many thanks,
> >
> > Authors

---

### Meta-Review · Area_Chair_sgHA · 2022-08-31

**Recommendation:** Reject
**Confidence:** Certain

**Metareview:**

The paper is interested to the sign language production (SLP) problem. A vector quantized conditional diffusion model is proposed for the pose generation. The proposed method achieves state-of-the-art results on the SLP evaluation benchmark (PHOENIX dataset).

Reviewers all agree that the key contribution is very interesting --  making VQ-diffusion work on SLP.  However, the technical novelty is low for NeurIPS. In that respect, reviewers agree that given VQ-diffusion has been shown (in past work) to perform quite well on text-to-image generation, the technical novelty is low here.

Several reviewers felt also the experimental section is a bit weak, mostly because PHOENIX is the only available benchmark for SLP.

Most other concerns have been fixed during the rebuttal phase. Following reviewers, the rejection is based on the limited technical novelty for NeurIPS.

**Award:**

No

---

### Decision · Program_Chairs · 2022-09-14

Reject